# GaN/Ga$_2$O$_3$ Core/Shell Nanowires Growth: Towards High Response Gas Sensors

**Quang Chieu Bui** [ID]**, Ludovic Largeau, Martina Morassi** [ID]**, Nikoletta Jegenyes, Olivia Mauguin, Laurent Travers, Xavier Lafosse, Christophe Dupuis, Jean-Christophe Harmand, Maria Tchernycheva and Noelle Gogneau** *

Centre de Nanosciences et de Nanotechnologies—CNRS-UMR9001, Université Paris-Sud, Université Paris-Saclay, 10 Boulevard Thomas Gobert, F-91120 Palaiseau, France
* Correspondence: noelle.gogneau@c2n.upsaclay.fr; Tel.: +33-(0)1-70-27-05-49



**Featured Application: GaN/Ga$_2$O$_3$ core/shell nanowires appear as promising solution to fabricate gas sensors working in a large range of temperature, from ambient temperature to high temperatures. Thanks to their core-shell heterostructures, these systems are based on the NW resistance modulation process, then offering the possibility to improve the sensor performances.**

**Abstract:** The development of sensors working in a large range of temperature is of crucial importance in areas such as monitoring of industrial processes or personal tracking using smart objects. Devices integrating GaN/Ga$_2$O$_3$ core/shell nanowires (NWs) are a promising solution for monitoring carbon monoxide (CO). Because the performances of sensors primarily depend on the material properties composing the active layer of the device, it is essential to control them and achieve material synthesis in the first time. In this work, we investigate the synthesis of GaN/Ga$_2$O$_3$ core-shell NWs with a special focus on the formation of the shell. The GaN NWs grown by plasma-assisted molecular beam epitaxy, are post-treated following thermal oxidation to form a Ga$_2$O$_3$-shell surrounding the GaN-core. We establish that the shell thickness can be modulated from 1 to 14 nm by changing the oxidation conditions and follows classical oxidation process: A first rapid oxide-shell growth, followed by a reduced but continuous oxide growth. We also discuss the impact of the atmosphere on the oxidation growth rate. By combining XRD-STEM and EDX analyses, we demonstrate that the oxide-shell is crystalline, presents the β-Ga$_2$O$_3$ phase, and is synthesized in an epitaxial relationship with the GaN-core.

**Keywords:** core/shell nanowires; GaN; Ga$_2$O$_3$; metal-oxide semiconductor; gas sensor devices

## 1. Introduction

The rapid development of smart objects has led to the increased interest of sensor technologies to collect and exchange sensed data in real-time. With the development of nanotechnologies, the sensor efficiency and accuracy have been improved leading to a large extension of their application domains [1,2]. Among a wide range of sensor applications, such as environmental monitoring or public safety, the development of sensor devices allowing monitoring of harmful greenhouse gases (GHGs) is of the utmost importance. GHGs need to be monitored in areas as diverse as the monitoring of industrial processes or personal tracking. Carbon monoxide (CO) is one of the GHGs. Its efficient detection and monitoring is essential and require the development of specific sensor devices, which must meet the following criteria: reliable, responsive, highly sensitive, miniaturized, low cost, and capable to operate on a large range of temperature, from ambient temperature (smart objects) to high temperatures (up to 600 °C for industrial combustion processes).

Most of actual sensors are solid-state sensors based on the interaction of gas particles with surfaces and volumes. The electrochemical potential, the resistivity, the density, and/or the optical properties are altered upon gas adsorption. This alteration must be maximized to increase the detection. Common high-temperature GHG sensors are based on metal oxide materials working on the principle of the chemiresistor. For CO, the detection mechanism is based on the redox reaction between the gas species and the pre-adsorbed $O_2$ on the surface of metal oxide, affecting the depletion layer and thus resulting in the change of the conductivity of the material [3–7].

Metal oxide sensors based on $SnO_2$ material have been developed to monitor CO due to their relatively high sensitivity [8,9]. Indeed, the detection of 5 ppm CO in few seconds at 300 °C has been demonstrated [10,11]. Chemiresistor sensors based on titanium oxide ($TiO_2$)/lanthanum oxide ($La_2O_3$/CuO) have also demonstrated their capacities to selectively detect down to 500 ppm of CO against $CH_4$ [12] under 5% of $O_2$ up to 600 °C. However, these kinds of sensors can present a low selectivity (due to interferences with other gases), a drift of the sensor response and/or a poor recovery. CO-sensors based on $Ga_2O_3$ thin films have also been developed, usually operating in the 400–600 °C temperature range. Comparing to $SnO_2$, $Ga_2O_3$ gas sensors present faster response, faster recovery time and lower cross-sensitivity to humidity [13–16]. The sensitivity and selectivity performances of these $Ga_2O_3$ sensors have been improved by adjusting the oxygen vacancy concentration [17] or by using gold particles to catalyze the reaction between adsorbed oxygen on sensor surface and gas molecules by reducing their activation energy [18]. However, these sensors based on planar films present numerous drawbacks. Among them, the limited surface, where takes place the interactions between gas molecules and materials, results in limited sensor performances.

These last decades, a new class of sensors based on 1D-nanstructures, such as nanowires (NWs) or nanorods (NRs), has appeared as a promising way to improve the sensing performances. Thanks to their specific properties, these 1D-nanostructures present, in comparison with their 2D-film counterparts, important characteristics to fundamentally improve the gas-sensing efficiency in terms of sensitivity, response and spatial resolution [19–28]: (i) their large surface to volume ratio greatly enhances the detection limit, thus leading to a higher sensitivity [29] and performances [30], and induces enhanced and tunable surface reactivity implying possible room temperature operation [31,32], faster response and recovery time [30]; (ii) their nanometer scale dimensions are compatible with the size of species being sensed, opening up the way for nano-detection; and finally (iii) due to these morphological properties combined with their quasi-lattice perfection, the NWs offer superior mechanical properties leading to large elastic deformation without plastic deformations or fractures [33], higher flexibility, higher robustness and higher resistance to fatigue, then extending the operational lifetime of nano-systems [34].

As for 2D based sensors, the main materials used for developing gas-sensors, from room temperature to 400 °C, are chemiresistor based on metal oxide NWs such as $SnO_2$, ZnO, $In_2O_3$ and $Ga_2O_3$. However, in spite of the considerable efforts devoted to developing highly efficient gas sensors based on 1D-nanostructures working on a large temperature range, further improvements of the sensor performances are needed.

To enhance the sensing performances of NW-based sensors, several approaches have been considered to modify the surface properties, such as doping, surface functionalization or hybridization [23] and core-shell heterostructures [23,35]. Concerning this last approach, carbon nanotubes-core/vanadium oxide-shell [36], $Ga_2O_3$-core/$SnO_2$-shell NWs [37], ZnO-core/$Co_3O_4$-shell NWs [38], Pb nanocubes-core/$In_2O_3$ oxide-shell [39] and $Ga_2O_3$-core/GaN-shell NWs [40] have demonstrated enhanced sensing performances or sensing at a lower operating temperature. These systems are based on the NW resistance modulation process. For example, $Ga_2O_3$/GaN core/shell-based sensor has demonstrated detection of 10–200 ppm CO at 150 °C with responses 1.6–3.1 times stronger than a pure $Ga_2O_3$ nanowire sensor [40]. In spite of these encouraging results, the system presents a strong response to other gases, which indicates its poor selectivity to CO.

To develop highly sensitive and selective CO-gas sensor devices operating in a large range of temperature, from ambient temperature (smart object) to high temperatures (up to 600 °C for industrial

combustion processes), we propose to consider GaN/Ga$_2$O$_3$ core-shell NW heterostructures combining the advantages of the GaN-core and Ga$_2$O$_3$-shell materials. GaN and its alloys present favorable properties to develop gas-sensors and especially in playing the NW-core role: (i) the III-Nitrides present high thermal and chemical stability as well as radiation hardness, thus allowing their use in harsh environments; (ii) they exhibit less intrinsic leakages and are capable of operating at high temperatures due to their larger bandgap and stronger bond energy [19]; (iii) the GaN, as same as AlN, has a high Pauling electronegativity difference (1.23 eV), which confers a high sensitivity of the electronic properties to its surface states (K Parameter) [19]; and (iv) the nitrides are characterized by conductivity in the range of $10^{-8}$ to $10^3$ S/cm, allowing us to keep the influence of the surface charge effects on the charge carrier concentration and on the modulation of the Fermi level [19]. The choice of the Ga$_2$O$_3$-shell is motivated by the following advantages: (i) Ga$_2$O$_3$ presents high thermal stability, especially the β-Ga$_2$O$_3$ crystalline phase (the most stable phase), which is especially suitable for CO sensing in high temperature environments; (ii) the resistivity of Ga$_2$O$_3$ is very sensitive to CO at high temperature [41]; and (iii) the presence of Ga$_2$O$_3$ layer covering the GaN surface increases the response of GaN sensors to CO [19].

The mechanism of gas sensors integrating GaN/Ga$_2$O$_3$ core-shell NWs is based on the NW resistance modulation before and after exposition to gases, such as CO. The core-shell NW heterostructures are characterized by the existence of two depletion zones: the first one at the air/shell interface, and the second one at the shell/core interface [40]. The adsorption/desorption of the gas molecules on the heterostructure surface modifies the depth of the first depletion layer, which in turn alters the depth of the second one, and thus induces a modulation of the corresponding potential barrier heights. Carrier transport in the core is thus affected, resulting in a large change in resistance and thus in an enhanced response of the NW-based sensor system.

Because the performance of sensors primarily depends on the material properties composing the active layer of the device, it is of crucial importance to control these properties and achieve material synthesis in the first time. In this paper, we investigate the synthesis of GaN/Ga$_2$O$_3$ core-shell NWs with a special focus on the formation of the Ga$_2$O$_3$ shell. The GaN NWs grown by plasma-assisted molecular beam epitaxy (PA-MBE), are post-treated following thermal oxidation to form Ga$_2$O$_3$-shell surrounding the GaN-core. We establish that the thickness of the Ga$_2$O$_3$-shell can be modulated by changing the oxidation conditions (temperature, O$_2$ flux and time of oxidation), from 1 to 14 nm. By investigating the thickening of the shell, we establish that the oxidation process follows the classical model with an initial rapid oxide-shell growth, followed by a second stage of reduced but continuous oxide growth. We also discuss the influence of the phase environment on the oxidation rate. Finally, the structural properties of the shell are investigated by combining XRD, TEM-STEM and EDX analyses. Then, we demonstrate that the oxide shell is crystalline, presents the β-Ga$_2$O$_3$ phase, the most stable one, and is synthesized in epitaxial relationship with the GaN-core.

## 2. Materials and Methods

The synthesis of GaN/Ga$_2$O$_3$ core-shell NW heterostructures is composed of two successive steps:

1-GaN-core NW growth: Self-assembled free-catalyst GaN-core NWs were grown on conductive oxide-free Si (111) substrate (resistivity of the order of 0.007 Ω cm) in a molecular beam epitaxy (MBE) chamber (RIBER Compact 12 PA-MBE system, Bezons, France), equipped with a radio-frequency N plasma source. After a chemical cleaning and a thermal deoxidation of the substrate surface to remove the organic pollutants and the native oxide, a 2.5-nm-thick AlN buffer layer was deposited at 620 °C following a previously reported procedure [42]. This seed layer allows a better control of the NW nucleation and density, a reduction of the NW twist and an improvement of the NW vertical orientation [42–46]. Then, the temperature was ramped to 790 °C to grow the GaN NWs under nominally N-rich conditions with an N/Ga flux ratio of 1.36. Following this growth procedure, the GaN NWs are vertically oriented with a hexagonal shape delimited by {10-10} plane, are characterized by a quasi-crystalline perfection (absence of dislocations) [46], a prevalent N-polarity [42,47] and

present reproducible dimensions and densities (Figure 1). GaN NWs considered in this study are characterized by a density of $5 \cdot 10^9$ NW/cm$^2$, an average diameter of 40 ± 5 nm and an average height of 650 ± 100 nm, these structural characteristics being extracted from high-resolved SEM images (plan-view and cross-section view).

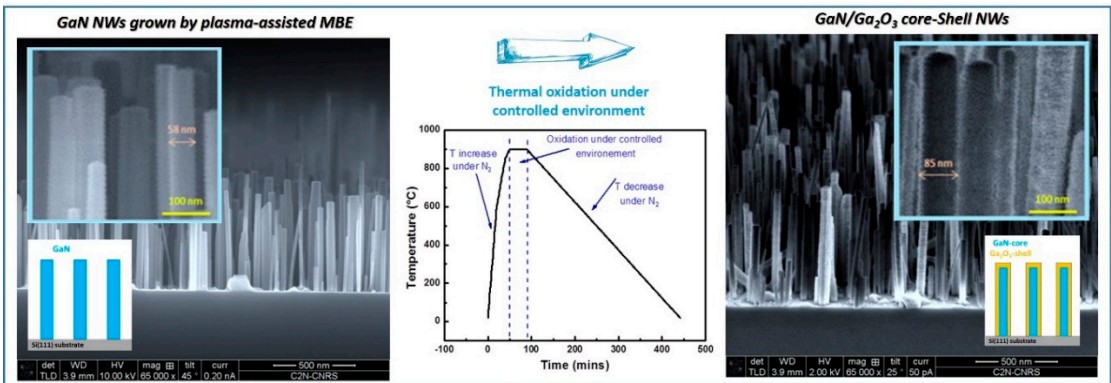

**Figure 1.** Schematization and SEM images of GaN nanowires (NWs) before oxidation and GaN/Ga$_2$O$_3$ core-shell NWs after oxidation. The cycle of thermal oxidation is also schematized.

2-Ga$_2$O$_3$ shell synthesis: The Ga$_2$O$_3$-shell was formed around GaN NW by thermal oxidation post-treatment. The as-grown GaN NWs were oxidized at high temperature following the procedure described in Figure 1. The adjustment of the oxidation temperature was performed under Nitrogen to avoid any alteration of the NW surfaces. After the stabilization of the targeted oxidation temperature, the controlled environment was injected, and the oxidation time was considered. Finally, the temperature was decreased to the ambient one under Nitrogen. During the oxidation step, due to the high temperature, the GaN is decomposed at the surface. The Nitrogen is removed in the environment, while the Ga, with its free dangling bonds, is linked with oxygen, then forming Ga$_2$O$_3$ oxide (Figure 1). After the formation of the first oxide monolayer, the oxide shell plays the role of a barrier separating the oxygen-based environment and the GaN NWs. To continue further the oxidation, the oxygen species should diffuse through the Ga$_2$O$_3$ oxide to form an additional oxide monolayer. As a function of the oxidation time, the shell surrounding the GaN-core becomes thicker.

The oxidation mechanisms, the rate of oxidation, the quality of Ga$_2$O$_3$ layer and the GaN/Ga$_2$O$_3$ interface can be affected by many factors such as temperature, oxidized agents and their flow rate, pressure, GaN structure and the concentration of impurities. In order to investigate the formation of the Ga$_2$O$_3$-shell, the oxidation of the GaN-core NWs has been tested under different conditions summarized in Table 1, such as oxidation time, temperature, N$_2$/O$_2$ ratio and humid or dry atmosphere. We note here that the N$_2$/O$_2$ ratio was chosen to be approximately equivalent to the normal air composition. The "humid" term refers to oxidation environment equivalent to normal air (i.e., with water vapor), while the "dry" term refers to the same oxidation conditions (N$_2$/O$_2$ ratio) but without water vapor.

**Table 1.** Conditions of oxidation.

| Factor<br>Series | Temperature (°C) | O$_2$ Flux (sccm) | N$_2$ Flux (sccm) | Time (min) |
|---|---|---|---|---|
| 1 | 900 | 1.3 dry atmosphere | 5 | 5–15 |
| 2 | 900 | 1.3 humid atmosphere | 5 | 3–15 |
| 3 | 850 | 1.3 dry atmosphere | 5 | 7–30 |
| 4 | 850 | 0.4–1.3 dry atmosphere | 5 | 15 |

The structural properties of the core-shell NWs have been characterized by combining different techniques of characterization. To evaluate the shell formation, the diameters of NWs were

systematically characterized before and after each oxidation step by high resolution scanning electron microscope (SEM, Magellan 400L-FEI, Hillsboro, OR, USA) in cross-section view and confirmed by transmission electron microscope (TEM, FEI, Hillsboro, OR, USA) and Energy Dispersive X-ray (EDX, FEI, Hillsboro, OR, USA) analyses. The crystal structure of NWs was analyzed by X-Ray diffraction (XRD, Smartlab RIGAKU, Tokyo, Japan) under in-plane diffraction configuration. The choice of this experimental configuration is motivated by the thinness of the sample (the average height of the NWs being of 650 nm) and by the fact that in our vertically oriented NWs configuration, the plans analyzed in this study are parallel to the c-axis. Finally, the composition and crystal structure of samples after oxidation were analyzed by using highly resolved TEM and scanning TEM modes.

## 3. Results and Discussion

In order to adjust the performances of sensors based on GaN/Ga$_2$O$_3$ core-shell NW, it is of crucial importance to understand the conditions of the shell formation and thus to control its thickness and crystalline quality. The main factors driving the thermal oxidation of GaN NWs are the temperature, the O$_2$ flux, the time and the dry/humid environment. The effects of these factors were studied through different series of samples (see Table 1). To thoroughly investigate the formation of the oxide shell, we have combined various techniques of characterization allowing us to access to the morphology modification of the NWs, as well as the composition and crystallographic phase of the oxide shell.

### 3.1. Ga$_2$O$_3$-Shell Thickness Evolution and Formation

By comparing the high-resolved SEM images of NWs before and after the oxidation, we can observe that the as-grown MBE GaN-core NWs are characterized by smooth surfaces, while the GaN/Ga$_2$O$_3$ core-shell NW heterostructures present rough surfaces (Figure 1). The formation of Ga$_2$O$_3$ shell is also characterized by an increase of the NW diameter.

Due to the self-assembled growth mode of the NWs (cf. Materials and Methods), we have a modulation of the as-grown NW diameter from wire to wire. To realize a representative study, each sample has been measured before and after oxidation, in a statistical view with the measurement of a large number of nanostructures (around 400 NWs). The diameter follows a single Gaussian distribution, within the limits of statistical error where the error bars correspond to the full width at half maximum of the Gaussian fit in each case (Figure 2a). The shell thickness is estimated from the difference between the mean values of the two distributions (before and after oxidation) with standard error calculated as follows: $\sigma_{2-1} = \sqrt{\frac{\sigma_1^2}{N_1} + \frac{\sigma_2^2}{N_2}}$ .

The diameter expansion is caused by the difference between the specific volume of the GaN (13.5 cm$^3$/mol) and the β-Ga$_2$O$_3$ (31.8 cm$^3$/mol), calculated based on their molar masses and densities (for GaN: 83.7 g/mol and 6.2 g/cm$^3$ [48]; for β-Ga$_2$O$_3$ 187.4 g/mol and 5.9 g/cm$^3$ [49]). The specific volume of β-Ga$_2$O$_3$ being higher than the one of GaN, the final diameter of Ga$_2$O$_3$/GaN NWs is higher than the initial one of GaN NWs. We want to note that even if the formation of the oxide mainly explains the expansion of the NW diameter, other reasons can accompany this one, such as the defects of Ga$_2$O$_3$ structure or the vacancies in the shell. It should also be noted that the difference between the mean values of the two distributions before and after oxidation is the average expansion of the NW diameter. However, it is not exactly equivalent to two times the thickness of the formed oxide. In fact, when GaN is oxidized, GaN layers are converted into the oxide layer. This means that the oxide thickness is equal to the thickness expansion plus the thickness of GaN layers that have been oxidized, as schematized on Figure 2b.

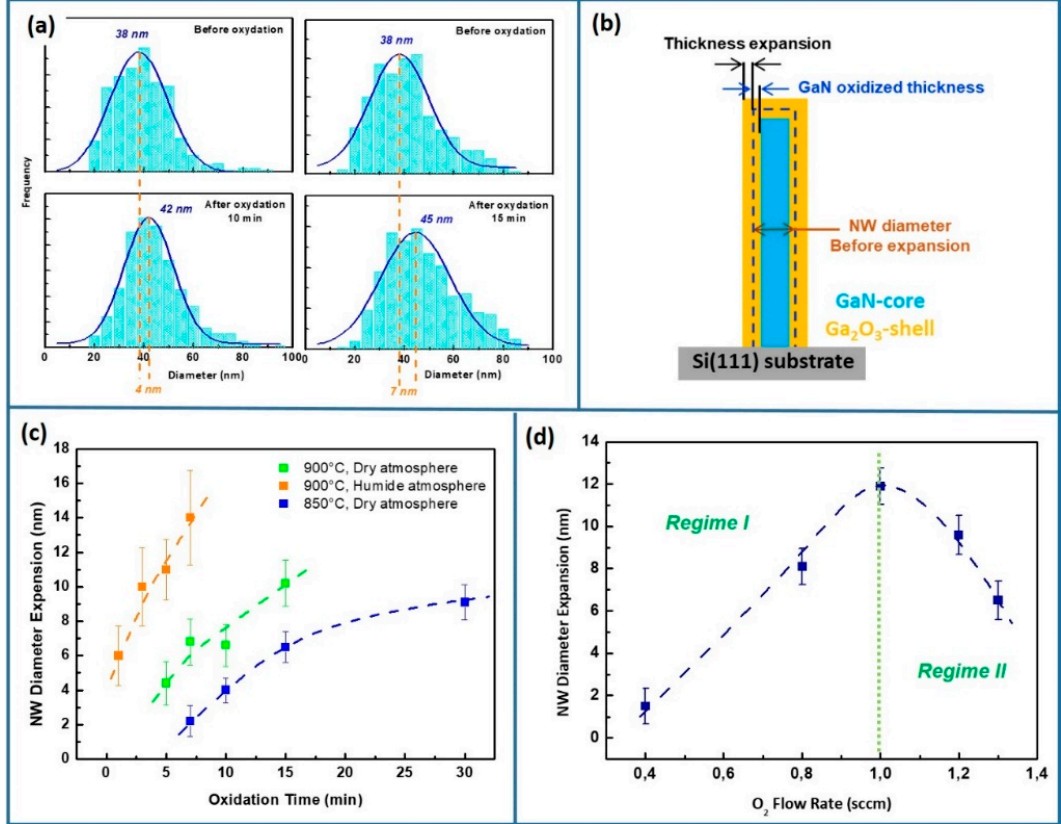

**Figure 2.** (**a**) Gaussian distribution of the NW diameter before and after oxidation, during 10 min and 15 min under a dry atmosphere at 850 °C for 1.3 sccm of $O_2$ and 5 sccm of $N_2$. The diameter expansion is schematized with dashed lines; (**b**) Schematization of the NW expansion; (**c**) NW diameter expansion as a function of the oxidation time for 1.3 sccm of $O_2$ and 5 sccm of $N_2$, for different temperatures and atmospheres; (**d**) NW diameter expansion as a function of the $O_2$ flux for a fixed temperature of 850 °C, a $N_2$ flux of 5 sccm, an oxidation time of 15 min and under dry atmosphere.

Figure 2c,d establish the direct correlation between the diameter expansion and the oxidation conditions, with an evolution of the $Ga_2O_3$ shell thickness between 1 and 14 nm. Figure 2c shows the diameter expansion at 850 °C (blue curve) and 900 °C (green curve) as a function of the oxidation time, for $O_2$ and $N_2$ fluxes of 1.3 sccm and 5 sccm, respectively, and under dry atmosphere (Table 1, Series 1 and 3). The shell thickness increases with the time and the temperature. This behavior is similar to the one observed for oxidized GaN or AlN films [50–54]. For the lower temperature (850 °C), two different oxidation processes can be distinguished: an initial rapid oxide-shell growth, followed by a second stage of reduced but continuous oxide growth (after about 15–20 min of oxidation). For the higher temperature (900 °C), the diameter expansion is more pronounced and can only be described by the first oxidation process, then demonstrating that the initial oxidation stage is dependent on the temperature. These descriptions of the oxidation process are consistent with the ones theoretically predicted [55] and experimentally observed [51,54,56]. Then, the oxidation process is characterized by two different regimes: The interfacial reaction-controlled and the oxidation diffusion-controlled regimes. In interfacial reaction-controlled regime, the GaN oxidation process is controlled by the surface reaction rate, leading to a linear oxidation [57]. As the oxidation duration is important, as the oxide thickness increases. This process is also temperature dependent, as experimentally observed (Figure 2c). The increase of the oxidation rate with the temperature results from the increase of the $O_2$ sticking probability attributed to a higher activation energy for $O_2$ dissociation [54,58]. In an oxidation diffusion-controlled regime, the oxidation process is controlled by the reagent transport through the growing gallium oxide layer. The oxidation kinetic can thus be described by a parabolic

model [57]. Although this regime is note observed for oxidation at 900 °C (green curve), we can assume an under-linear thickening of the shell for higher oxidation times.

Figure 2c also presents the diameter expansion as a function of the time for the same oxidation conditions (T = 900 °C, $O_2$ = 1.3 sccm, $N_2$ = 5 sccm), but for different atmospheres, i.e., under dry (green curve) and wet (orange curve) thermal oxidation conditions. We can clearly show that the diameter expansion of oxidized NWs is more important under wet atmosphere. In other words, under wet atmosphere, the oxidation rate is much pronounced in comparison with the one occurring under dry atmosphere. This result points to the importance of the oxidation atmosphere composition on the GaN oxidation process. To understand the mechanisms in play, we need to take into account the different physical and chemical processes occurring during the oxidation step. In this experiment (Figure 2c), the oxidation being carried out at the same temperature of 900 °C and with equivalent oxygen flow, we can assume that the gas diffusion rate in the gas phase and at the gas/oxide interface are constant and do not influence the GaN oxidation. However, by regarding the atmosphere composition, the surface processes (chemisorption, decomposition and diffusion through the oxide layer) are not equivalent. Once chemisorbed at the NW surface, the reactive species are decomposed with the following equations:

$$\text{Under dry environment: } 4GaN_{(s)} + 3O_{2(g)} \rightarrow 2Ga_2O_{3(s)} + 4N_{2(g)} \tag{1}$$

$$\text{Under wet environment: } 2GaN_{(s)} + 3H_2O_{(g)} \rightarrow 2Ga_2O_{3(s)} + 2N_{2(g)} + 3H_{2(g)} \tag{2}$$

The species migrating at the surface, at the metal-oxide interface, and through the oxide layer are ionic species. Under a dry environment, the chemisorbed oxygen is dissociated to an ionic form of $O^{2-}$. Under wet environment, the oxygen coming from the chemisorbed $H_2O$ is dissociated to an ionic form of $OH^-$. After the formation of the first oxide layer covering the GaN-core NW (Figure 3a,c), the active species must to diffuse through the oxide film in order to form the $Ga_2O_3$. The diffusion rate of $O^{2-}$ and $OH^-$ through the oxide layer being more efficient that the one of metallic ionic species ($Ga^{2+}$) [54,57,59], the diffusion process is controlled by the oxygen-based active species. Thus, the $O^{2-}$ and $OH^-$ are diffusing through the oxide layer and are combined with the $Ga^{2+}$ species diffusing at the metal-oxide interface to form the $Ga_2O_3$-shell (Figure 3b,d). The substituted nitrogens (resulting from the GaN decomposition) by oxygen diffuse through the oxide layer towards the oxide/gas interface, being thus able to lead to the formation of interstitial nitrogen [57]. The fact that oxidation rate is higher under wet environment in comparison with dry environment (Figure 2c, green and orange curves), results from the ability of $OH^-$ to diffuse more efficiently through the already-grown oxide layer than the $O^{2-}$ [57]. The difference in diffusion rate is also accentuated by the porosity of the oxide layer form under wet environment, due to the formation of nitrogen and hydrogen during the oxidation reaction [57,60].

The porosity of the oxide layer is associated with a poor $GaN/Ga_2O_3$ interface causing unstable electrical properties of the structures [15]. By regarding the degradations of the electrical properties, and the faster oxidation yielding to a more difficult control over the oxide formation, the wet oxidation conditions can be considered as inappropriate to develop gas-sensors with high performances.

Finally, we have analyzed the oxidation under dry atmosphere as a function of the $O_2$ flux (varying between 0.4 and 1.3 sccm), by keeping constant the temperature at 850 °C, the $N_2$ flux at 5 sccm and the oxidation time at 15 min (Figure 2d). We can observe two different regimes of oxidation: Regime I is characterized by a continuous increase of the oxidation rate with the $O_2$ flux, while in Regime II, the oxidation rate decreases for high fluxes. To understand the origin of these two regimes, we have to consider that the variation of oxygen flow rate can affect many factors during oxidation, such as the concentration of oxygen and nitrogen, the total pressure in the oxidation chamber, as well as the partial pressure of oxygen and nitrogen.

The nitrogen is an inert gas, so that it is not participating to the oxidation reaction. By contrast, the increase of the oxygen flow induces an increase of the oxygen concentration in the $N_2/O_2$ gas phase and

thus an increase of the chemisorbed gas molecules at the surface. The oxidation rate being controlled by the number of gas molecules diffusing through the oxide shell, the enhancement of the oxygen flow is associated with an increase of the shell-thickness growth rate. Moreover, from the GaN decomposition point of view, when the sample is oxidized in environment with low oxygen flow rate, partial pressure of nitrogen is high, which can reduce the decomposition of GaN as mentioned in report [14]. In other words, the decomposition of GaN gets faster as the oxygen flow rate becomes higher and the partial pressure of nitrogen becomes lower. These behaviors are both in good agreement with the diameter expansion observed in Regime I, where the oxygen flow rate increases from 0.4 to 1.0 sccm. However, for fluxes higher than 1.0 sccm, the diameter expansion decreases. Similar effect has been observed for 4H-SiC MOS capacitors, where the oxide thickness is reduced by increasing the $O_2$ flow rate [61]. This behavior can appear contradictory to the last one described. To explain the behavior in Regime II, the increase of the total pressure with the oxygen flow rate can be considered. This means that the GaN decomposition could be more difficult under high pressure, resulting in a slower oxidation rate. Additional experiments are needed to understand this Regime II. Nevertheless, the less pronounced oxidation rate for the high $O_2$ fluxes does not prevent the control of the $Ga_2O_3$-shell thickness from 1 to 14 nm by adjusting the oxidation temperature, time and atmosphere, as illustrated by Figure 2c, where the $O_2$ flux was fixed to 1.3 sccm, the highest tested one.

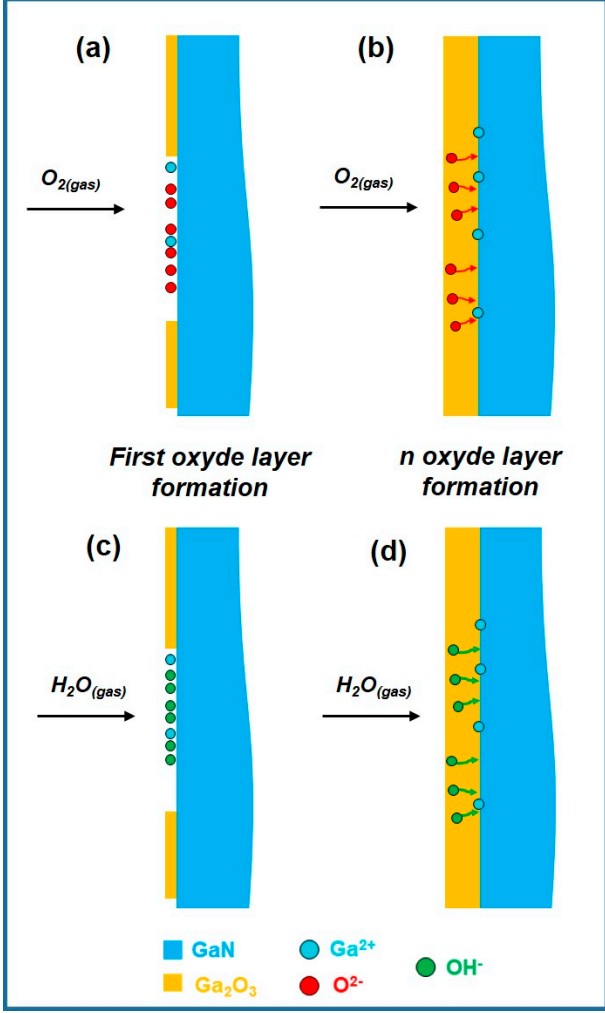

**Figure 3.** Schematization of the oxidation processes of GaN-core NW under dry (**a**,**b**) and humid (**c**,**d**) atmospheres.

### 3.2. Structural Characterization of the Ga$_2$O$_3$-Shell

　　To study the composition and the crystal structure of the Ga$_2$O$_3$ oxide shell, samples have been characterized by combining XRD, STEM and EDX analyses. Figure 4 shows typical spectra of GaN NWs before oxidation and GaN/Ga$_2$O$_3$ NWs after oxidation, analyzed by XRD using in-plane Phi/2ThetaChi scan configuration. The spectra performed on as-grown GaN NWs (Figure 4a) shows only two peaks located at 47.3° and at 57.8° corresponding respectively to the signals of Si (220) and hexagonal GaN (11-20). By contrast, the spectra of NWs after oxidation (Figure 4c) presents two additional peaks, located at 61.1° and 64.4°, suggesting the signatures of oxide. According to reports [51,52], monoclinic β-Ga$_2$O$_3$ phase, the most stable one, appears after similar oxidation processes of GaN. By comparing with the reference XRD spectra of β-Ga$_2$O$_3$ shown in Figure 4b [62], the peak at 61.1° is attributed to the signal of β-Ga$_2$O$_3$ (020), as confirmed by STEM analyses (presented later), while the one around 64° is assigned to the signal of β-Ga$_2$O$_3$ (403).

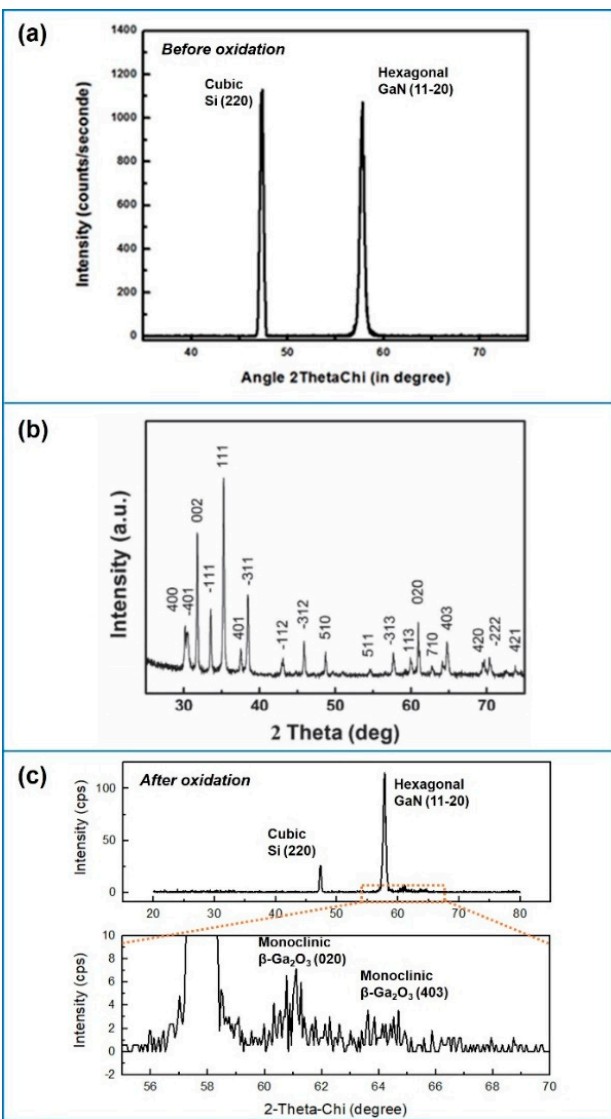

**Figure 4.** XRD analyses using in-plane Phi/2ThetaChi scan configuration. (**a**) XRD spectra of as-grown GaN NWs before thermal oxidation; (**b**) Reference XRD spectrum of the β-Ga$_2$O$_3$ in report [62] (Used with permission from JCPDS—International Centre for Diffraction Data); (**c**) XRD spectra of GaN/Ga$_2$O$_3$ NWs after oxidation at 900 °C during 7 min under dry atmosphere with N$_2$ and O$_2$ fluxes of 5 sccm and 1.3 sccm, respectively.

We note here that only the GaN (11-20) peak appears on Figure 4a,c. The peak of GaN (10-10) at 32.3903° does not appear, despite it is perpendicular to the sample surface and thus can give signal in XRD in-plane scanning configuration. However, in XRD phi/2thetachi scanning mode, to observe several signatures in the same spectrum, the planes must be parallel to each other. The (11–20) plane cannot be parallel to the (10-10) one in same NWs as the 1D-nanostructures grown by PA-MBE are monocrystalline with a unique orientation. The missing of GaN (10-10) peak in Figure 4 confirms that NWs are preferentially oriented, as previously demonstrated [46].

The detailed positions of peaks for three different samples before and after dry oxidation are showed in the Table 2. The theoretical values of 2ThetaChi have been calculated by using theoretical parameters of crystal.

**Table 2.** Peak positions for three different samples before and after dry oxidation.

| | Si (220) [1] | GaN (11-20) | $Ga_2O_3$ (020) | $Ga_2O_3$ (403) |
|---|---|---|---|---|
| T = 900 °C, $N_2/O_2$ = 5/1.3, Time = 7 min, Shell thickness = 6.9 | 47.3516° | 57.8236° | 61.0465° | 64.6155° |
| T = 850 °C, $N_2/O_2$ = 5/1.3, Time = 15 min, Shell thickness = 6.5 | 47.3277° | 57.8005° | 61.0825° | 64.0635° |
| T = 850 °C, $N_2/O_2$ = 5/0.8, Time = 15 min, Shell thickness = 8.1 | 47.3262° | 57.8005° | 61.0400° | 64.4000° |
| Pure GaN NWs | 47.3211° | 57.7507° | | |
| Theory 2ThetaChi | 47.3010° | 57.7742° | 60.8969° | 64.6319° |

[1] Because the crystal parameters remain unchanged after oxidation process, the Si peak is considered to compare values of GaN and $Ga_2O_3$ in the different samples.

The mismatch existing between GaN and $Ga_2O_3$ at the core-shell interface induces internal strain and thus affects their parameters as well as values of 2ThetaChi in XRD spectrum. Due to the specific volume of GaN and $Ga_2O_3$, from a theoretical point of view, $Ga_2O_3$ should induce stretching in the GaN, and reciprocally, GaN should induce compression on $Ga_2O_3$. This strain equilibrium depends on the thickness ratio between the core and the shell. By regarding the thickness of the GaN-core (diameter ~40 nm) and of the oxide shell (for thickness < 10 nm), the main stress expressed in the heterostructure is localized in the shell, as demonstrated by the material crystal parameters extracted from XRD measurements, summarized in the Table 3. The average parameter of GaN-core does not changed while the increase of 2ThetaChi of $Ga_2O_3$ (020) in the Table 2 and the decrease of its parameter b, in Table 3, indicate the presence of a compressive $Ga_2O_3$ shell. For the specific case of thick $Ga_2O_3$ shell (thickness > 10 nm), the compressive strain starts to be relaxed. In consequences, this partially relaxed shell induces tensile strain on the GaN-core.

**Table 3.** Parameters of crystals extracted from XRD measurements.

| | a of Si | a of GaN | b of $Ga_2O_3$ |
|---|---|---|---|
| T = 900 °C, $N_2/O_2$ = 5/1.3, Time = 7 min, Shell thickness = 6.9 | 5.426 Å | 3.186 Å | 3.033 Å |
| T = 850 °C, $N_2/O_2$ = 5/1.3, Time = 15 min, Shell thickness = 6.5 | 5.428 Å | 3.188 Å | 3.031 Å |
| T = 850 °C, $N_2/O_2$ = 5/0.8, Time = 15 min, Shell thickness = 8.1 | 5.428 Å | 3.188 Å | 3.034 Å |
| Pure GaN NWs | 5.429 Å | 3.190 Å | |
| Theory parameter | 5.431 Å | 3.189 Å | 3.040 Å |

As shown on Figure 4, the peak of GaN (11-20) is very sharp, indicating uniform crystal parameter of GaN synthesized by PA-MBE. By contrast, the $Ga_2O_3$ (020) peak is quite broad and slightly shifted towards higher degrees. To explain these observations, we have to consider two phenomena: (1) due to the stress undergone from GaN and/or due to the possible presence of defects at the GaN/$Ga_2O_3$ interface, the crystal parameters of $Ga_2O_3$ are varied in whole oxide shell. The variation of crystal parameters leads to the variation of 2ThetaChi, causing the wide peak in XRD spectrum. (2) If we finely

analyze the reference XRD spectrum of β-$Ga_2O_3$ (Figure 4b), we can note the presence, very close to the $Ga_2O_3$ (020) peak, of the $Ga_2O_3$ (710) peak, with an estimated (from theoretical parameter) 2ThetaChi of 62.6294°. The widening of the $Ga_2O_3$ (020) and $Ga_2O_3$ (710) peaks due to the non-homogeneous crystallinity of the shell can lead to their overlap, then resulting in the appearance, on the XRD spectrum, of a unique $Ga_2O_3$ (020) broad peak slightly shifted with respect to the peak theoretical position.

XRD analyses evidence that the thermal treatment of MBE-grown GaN NWs leads to the formation of oxide-shell with β-$Ga_2O_3$ crystalline phase. To further investigate the formation of this shell, we have analyzed samples by XRD using in-plane Phi scan configuration. In this configuration, the samples are rotated by 360°, while the angle between the X-ray source and the detector (2ThetaChi) is fixed at 57.8°, 61.1° and 64.3° corresponding respectively to the GaN (11-20), β-$Ga_2O_3$ (020) and β-$Ga_2O_3$ (403) peak positions. Figure 5 presents the results for pure GaN NWs and GaN/$Ga_2O_3$ NWs. On Figure 5a, six peaks correspond to the six (11-20) planes of GaN, demonstrating the hexagonal symmetry of GaN. Figure 5b,c show also six peaks, indicating that the (020) and (403) planes of β-$Ga_2O_3$ present symmetries along the hexagonal structure of GaN. These results, in agreement with the result from Phi/2ThetaChi scan (Figure 4), demonstrate that the β-$Ga_2O_3$ shell synthesized by post-thermal treatment is grown in an epitaxial relationship with the GaN core.

The XRD analyses have evidenced that the $Ga_2O_3$ shell crystallize in the β-$Ga_2O_3$, the most-stable phase, but is characterized by a lack of uniformity. To investigate this point, we have performed highly resolved TEM and STEM analyses combined with EDX mappings. Figure 6a,c presents bright field and dark field images of GaN/β-$Ga_2O_3$ NWs. The crystal structure of the heterostructures is characterized by different patterns, corresponding to the structure of GaN for the core, and the structure of $Ga_2O_3$ oxide for the shell. The TEM images evidence a ripple at the GaN/oxide interface and show variations in the shell lattice. This could be caused by the mismatch between GaN and $Ga_2O_3$ structures, creating internal stress located in the shell as shown by the XRD analyses (Figure 4), and the appearance of defects in $Ga_2O_3$.

The zoom in the NW core (Figure 6c) indicates that the pattern responds to the (11-20) plane of hexagonal GaN. The perfect lattice of GaN in the figure indicates that its monocrystalline structure is preserved after the oxidation treatment. Concerning the shell, the TEM images reveal that it presents a different crystalline structure from the NW core and is non-uniform with different patterns. According to the result from thr XRD analyses, the (11-20) plane of GaN is parallel to the (020) one of β-$Ga_2O_3$. This means that the crystal structure of this last plane, or of any other parallel ones, can be observed in the same images as the one containing the (11-20) plane of GaN. Figure 6d presents STEM analysis performed at the GaN/$Ga_2O_3$ interface. The shell presents the same structure as the structure of β-$Ga_2O_3$ (020) plane [63]. Combined with the fact that their parameters correspond to the theoretical ones, we can confirm that the oxide shell formed by thermal oxidation of MBE-grown GaN NWs presents the monoclinic β-$Ga_2O_3$ crystalline phase [63]. The non-uniformity of the shell results from the mismatch caused by hexagonal structure of GaN, which disarranges the formation of monoclinic structure of $Ga_2O_3$ at the angles of hexagon. Since $Ga_2O_3$ structure presents an epitaxial relation with GaN structure, the angles of hexagonal structure change the direction of monoclinic structure in an inappropriate way, leading to the observed mixing direction of $Ga_2O_3$ lattice.

Finally, the Figure 7 presents EDX mappings performed on GaN/β-$Ga_2O_3$ NWs. The different elements composing the heterostructure can be well distinguished between the GaN-core and the $Ga_2O_3$-shell. At this stage, it is important to note that blue dots of oxygen atoms can be seen in the GaN-core area. However, this is not the signature of oxygen incorporation inside the GaN-core. Because the $Ga_2O_3$-shell surrounds the GaN-core, during the EDX analyses, the electron beam is transmitted through the entire NWs (GaN plus $Ga_2O_3$), then causing an overlap of the core and shell EDX signatures. The observed blue dots of oxygen in the core area is thus a visual effect specific to the NW heterostructure configuration as we have already described in [64]. The EDX mappings reveal that the β-$Ga_2O_3$ shell surrounds continuously the GaN-core, which is of crucial importance to integrating these kinds of heterostructures in gas sensors. EDX mappings evidence also the border between the

core and the shell, which expresses the ripple interface of GaN/Ga$_2$O$_3$ similar to ones observed by STEM (Figure 6). The thickness of the oxide shell extracted from EDX mappings is around 3–5 nm, this measurement being in good agreement with our estimations performed from SEM analyses (Figure 2).

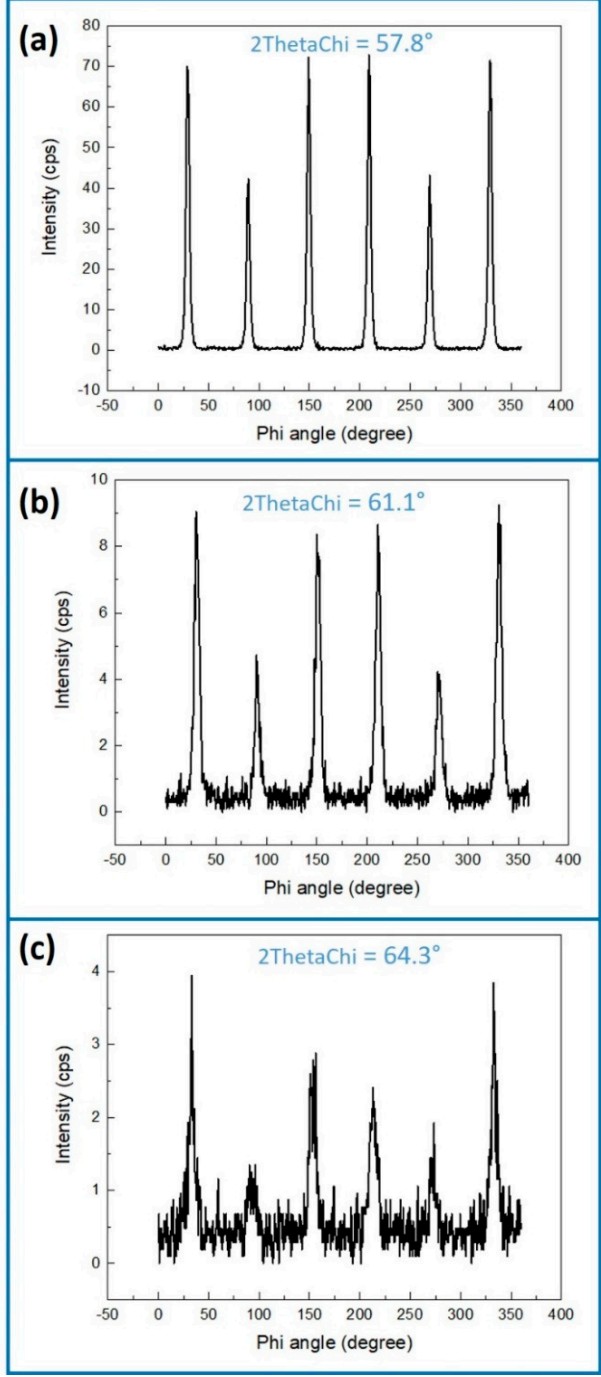

**Figure 5.** XRD analyses using in-plane Phi scan configuration of GaN/β-Ga$_2$O$_3$ NWs with the detector (2ThetaChi) fixed at 57.8° (**a**), 61.1° (**b**) and 64.3° (**c**) corresponding respectively to GaN (11-20), β-Ga$_2$O$_3$ (020) and β-Ga$_2$O$_3$ (403) peak positions.

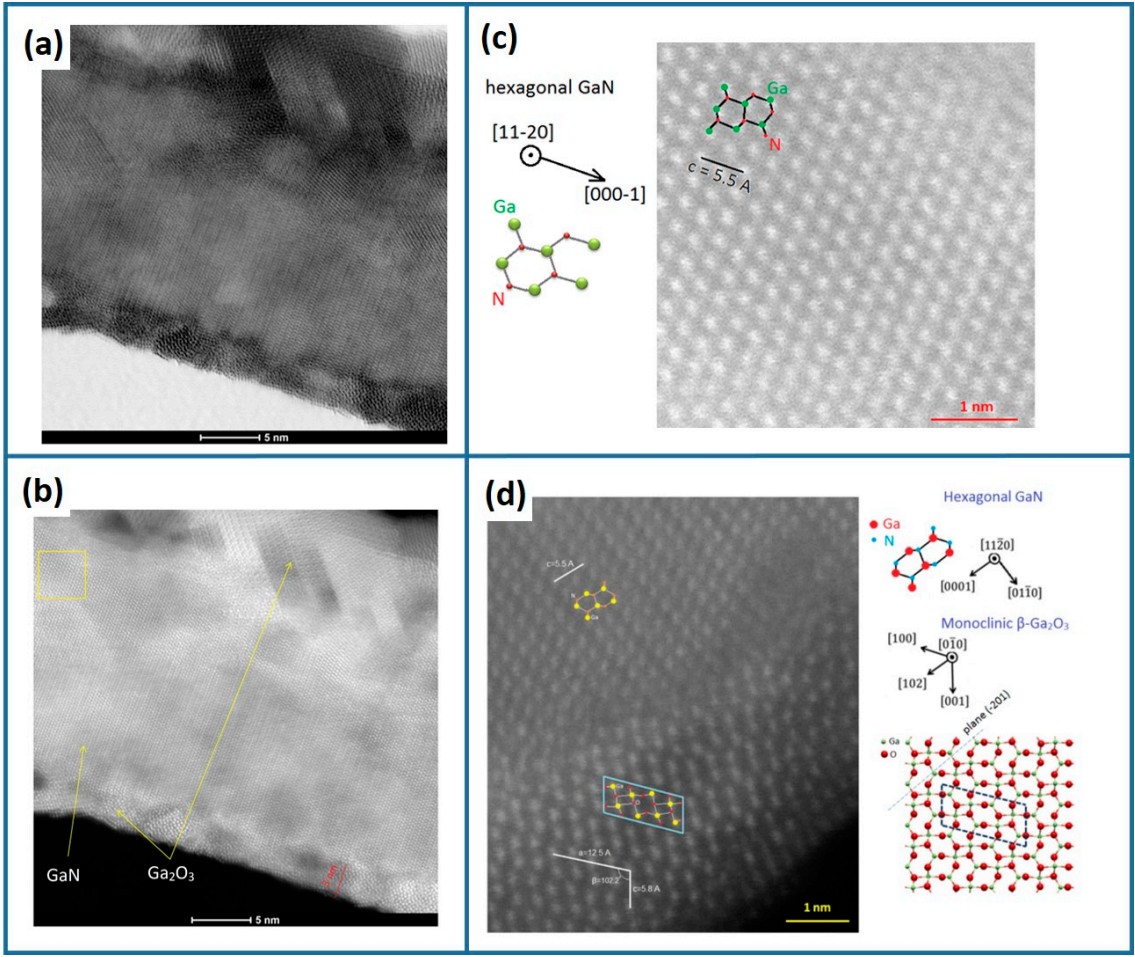

**Figure 6.** STEM images of GaN/Ga$_2$O$_3$ NWs in (**a**) bright field mode; (**b**) dark field mode. (**c**) Magnification of the GaN-core. (**d**) Magnification at the GaN/Ga$_2$O$_3$ interface. The β-Ga$_2$O$_3$ structural representation is extracted from [63].

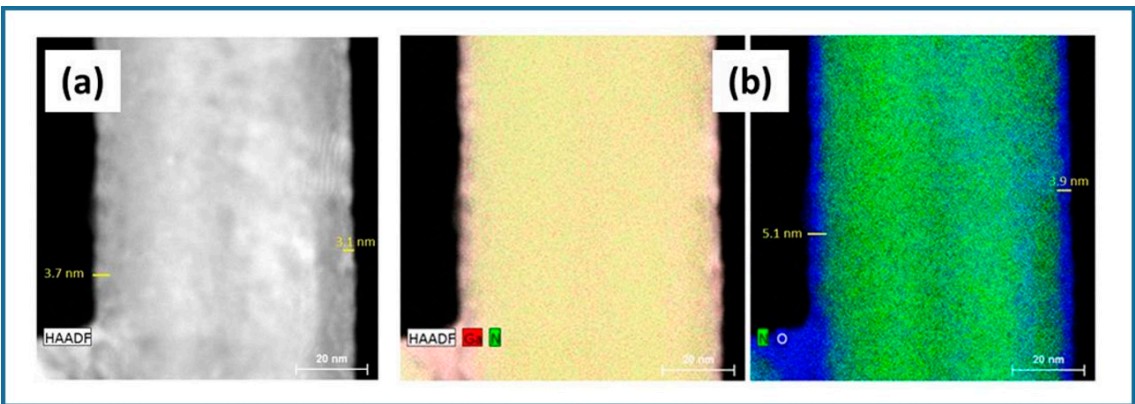

**Figure 7.** STEM (**a**) and EDX mappings (**b**) of GaN/Ga$_2$O$_3$ NW.

## 4. Conclusions

We have developed a synthesis procedure to form GaN/β-Ga$_2$O$_3$ NWs composed of two successive steps: (1) the growth of GaN NWs by plasma-assisted MBE, followed by (2) their thermal oxidation post-treatment, leading to the formation of a Ga$_2$O$_3$-shell surrounding the GaN-core. Due to the larger specific volume of Ga$_2$O$_3$ by regarding the one of GaN, the oxidation of the GaN-core is associated

with an expansion of its diameter. By analyzing the diameter expansion of the shell as a function of the oxidation conditions (temperature, $O_2$ flux, atmosphere and time of oxidation), we have established that the thickness of the $Ga_2O_3$-shell can be modulated with the oxidation conditions, from 1 to 14 nm. We have also shown that the oxidation process follows the classical model with an initial rapid oxide-shell growth, followed by a second stage of reduced but continuous oxidation rate. In the first regime, the GaN oxidation process is controlled by the surface reaction leading to a linear oxidation, while in the second regime, the oxidation process is controlled by the reagent transport through the growing gallium oxide layer. The influence of the oxidation environment on the oxidation rate has also been discussed. Then, the enhancement of the oxidation rate under a wet environment is associated with a faster diffusion through the oxide-shell of the $OH^-$ species, resulting from the oxygen decomposition under a wet atmosphere, than the $O^{2-}$ species, resulting from the $O_2$ decomposition under dry atmosphere.

The structural properties of the shell have been investigated by combining various techniques: XRD, TEM-STEM and EDX. We have demonstrated that the oxide shell formed by thermal oxidation is synthesized in epitaxial relationship with the GaN core, and presents the β-$Ga_2O_3$ crystalline phase, the most stable one.

By controlling the synthesis of GaN/β-$Ga_2O_3$ NWs, this paper constitutes a building block for developing high-efficient CO-gas sensors. GaN/β-$Ga_2O_3$ core-shell NW based sensor devices with different shell thicknesses (synthesis under dry and humid atmosphere) have been fabricated and are under investigation to establish the gas sensor performances as a function of the shell characteristics.

Finally, we would like to comment that GaN/β-$Ga_2O_3$ core-shell NW based sensors are promising for CO detection in harsh environments and under high temperatures. However, these nanostructures can also exhibit sensitivity to other gas at low temperature, such as oxygen [5] or hydrogen [65].

**Author Contributions:** NW growth, M.M., L.T., N.G., and Q.C.B.; Thermal oxidation, Q.C.B., X.L., and N.G.; Methodology, Q.C.B., M.M. and N.G.; NW structural characterizations, L.L., Q.C.B, C.D., O.M., and N.G..; Investigation, Q.C.B., L.L., and N.G.; Data Analysis, Q.C.B., M.M., N.J., L.L., M.T., and N.G.; Validation, M.T., and N.G.; Funding Acquisition, M.T., J.-C.H., and N.G.; Writing-Original Draft, Q.C.B., M.T., L.L., and N.G.; Supervision, M.T., J.-C.H., and N.G.

**Funding:** This work was financially supported by the French National Research Agency through the GANEX program (ANR-11-LABX-0014), and the EU Horizon 2020 ERC project 'NanoHarvest' (Grant 639052).

**Conflicts of Interest:** The authors declare no conflict of interest.

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
