# Peer review of "GaN/Ga2O3 Core/Shell Nanowires Growth: Towards High Response Gas Sensors"

_applsci, doi:10.3390/app9173528_

Round 1

Reviewer 1 Report

The manuscript dealt with GaN/Ga2O3 core-shell nanowires using post-treated following thermal oxidation towards high –efficient CO gas sensors. The authors have done a good job to understand structural properties of GaN/Ga2O3 using XRD, XRD-STEM and EDX analysis. However, some major criticisms have been made regarding this manuscript that should be corrected.

1.     GaN/Ga2O3 core/shell Nanowires growth: Towards high-efficient CO-sensors

This title is not recommended, Is GaN/G2O3 core-shell selective for CO gas? (If yes, No problem).

C.T. Lee et. Al. reported the hydrogen gas sensors (Sensors and Actuators B: 147 (2009) 723-729 and Z. Liu et al. reported Ga2O3 nanowire gas sensors, In which Ga2O3 nanowire exhibited high O2 response than CO gas.

Here my suggestion

2.      GaN/Ga2O3 core/shell Nanowires growth: Towards high response gas sensors

The authors mentioned that “the humid term refers to oxidation environment equivalent to normal air” authors should explain how humidity affect the oxidation in the manuscript?

3.     What is effect of Ga2O3 thickness on the gas sensing performance? Authors should comment on this issue!

4.      The technical writing is not sound for example,

In “High Resolution SEM images” H and R should be small, Energy Dispersive X-ray (EDX) written twice in the manuscript. Spelling mistakes (“developped” in the Conclusion), “This last decade”

5.  Recent applications of GaN nanorods for low temperature gas sensors should be added to the introduction such as following papers.

1. Sensors and Actuators B 264, (2018) 353-362,

https://www.sciencedirect.com/science/article/pii/S0925400518304969

2. Organic Electronics 65, 334-340 (2019)

https://www.sciencedirect.com/science/article/pii/S1566119918306244

3. Dalton Transactions, 48, 1367-1375 (2019) 

https://pubs.rsc.org/en/content/articlelanding/2019/dt/c8dt04709j#!divAbstract.

Author Response

Dear Editor,

We would like to thank each referee for his/her careful reading and remarks for our manuscript “GaN/Ga2O3 core/shell Nanowires growth: Towards high-efficient CO-sensors” by Q.C. Bui, et al. (manuscript ID: applsci-551370).

We have carefully considered each remark of the referees and have answered, modified, corrected and/or commented each one.

Please find below the referee’s comments immediately followed by our response. According to these comments, the manuscript has been revised, the text appearing in red corresponding to the added or modified informations.

All the modifications are appearing in red in the revised manuscript. With the reviewer’s comments, and the corresponding modifications, text adjustments have been performed, especially in the abstract, introduction and conclusion, to have a good coherence all along the manuscript. A new figure, as well as new references have been also added.

Thank you for your consideration.

Q.C. BUI and Dr. N. Gogneau

Reviewer 1:

The manuscript dealt with GaN/Ga2O3 core-shell nanowires using post-treated following thermal oxidation towards high –efficient CO gas sensors. The authors have done a good job to understand structural properties of GaN/Ga2O3 using XRD, XRD-STEM and EDX analysis. However, some major criticisms have been made regarding this manuscript that should be corrected.

Comment 1:  GaN/Ga2O3 core/shell Nanowires growth: Towards high-efficient CO-sensors

This title is not recommended, Is GaN/G2O3 core-shell selective for CO gas? (If yes, No problem). C.T. Lee et. Al. reported the hydrogen gas sensors (Sensors and Actuators B: 147 (2009) 723-729 and Z. Liu et al. reported Ga2O3 nanowire gas sensors, In which Ga2O3 nanowire exhibited high O2 response than CO gas.

Here my suggestion: GaN/Ga2O3 core/shell Nanowires growth: Towards high response gas sensors

We agree with the reviewers’ suggestion and we have modified the manuscript title as proposed. Thus, the new title of the paper is:

GaN/Ga2O3 core/shell Nanowires growth: Towards high response gas

         In addition, the two references proposed in this comment have been added in the conclusion of the manuscript to open the discussion as follow:

         Finally, we would like to comment that GaN/β-Ga2O3 core-shell NW based sensors are promising for CO detection in harsh environment and under high temperature. But, these nanostructures can also exhibit sensitivity to other gas at low temperature, such as oxygen [65] or hydrogen [66].

Comment 2:  The authors mentioned that “the humid term refers to oxidation environment equivalent to normal air” authors should explain how humidity affect the oxidation in the manuscript?

We agree that this part of the discussion is missing in the original manuscript.  We have thus modified completely the discussion of the Figure 2C, where the thickening of the shell is discussed as a function of the oxidation environment (dry and humid atmosphere), and thus include the following discussion:

Figure 2c also presents the diameter expansion as a function of the time for the same oxidation conditions (T = 900°C, O2 = 1.3 sccm, N2 = 5 sccm), but for different atmospheres, i.e. under dry (green curve) and wet (orange curve) thermal oxidation conditions. The comparison between samples shows that the diameter expansion of NWs oxidized in wet atmosphere is more important. In other words, in wet atmosphere, the oxidation rate is much pronounced in comparison with the one under dry atmosphere. These results point the importance of the oxidation atmosphere composition on the GaN oxidation process. To understand the mechanisms in play, we need to take into account the different physical and chemical processes occurring during the oxidation step. In this experiment (Figure 2c), the oxidation being carried out at a same temperature of 900°C and with equivalent oxygen flow, we can assume that the gas diffusion rate in the gas phase and at the gas/oxide interface are constant and do not influence the GaN oxidation. However, by regarding the atmosphere composition, the surface process (chemisorption, decomposition and diffusion through the oxide layer) are not equivalent. Once chemisorbed at the NW surface, the reactive species are decomposed with the following equations:

In dry environment:  4GaN(s) + 3O2(g) → 2Ga2O3(s) + 4N2(g)                              (1)

In wet environment:  2GaN(s) + 3H2O(g) → 2Ga2O3(s) + 2N2(g) + 3H2(g)          (2)

The species migrating at the surface, at the metal-oxide interface, and through the oxide layer are ionic species. In dry environment, the chemisorbed oxygen is dissociated to an ionic form of O2-. In wet environment, the oxygen coming from the chemisorbed H2O is dissociated to an ionic form of OH-. After the formation of the first oxide layer covering the GaN-core NW (Figures 3a and c), the active species must to diffuse through the oxide film in order to form the Ga2O3. The diffusion rate of O2- and OH- through oxide layer being more efficient that the one of metallic ionic species (Ga2+) [54, 58, 60], the oxygen-based active species are controlling the diffusion process. Thus, the O2- and OH- are diffusing through the oxide layer and are combined with the Ga2+ species diffusing at the metal-oxide interface to form the Ga2O3 shell (Figures 3b and d). The substituted nitrogen by oxygen diffuse through the oxide layer towards the oxide/gas interface, thus being able to lead to the formation of interstitial nitrogen [58].

The fact that oxidation rate is higher under wet environment in comparison with dry environment (Figure 2c, green and orange curves), results from the ability of OH- to diffuse more efficiently through the already-grown oxide than the O2- [58]. The different in diffusion rate is also accentuated by the porosity of the oxide layer form under wet environment, due to the formation of nitrogen and hydrogen during the oxidation reaction [58, 61]. The porosity of the oxide layer is associated with a poor GaN/ Ga2O3 interface causing unstable electrical properties of the structure [15]. By regarding the degradation of the electrical properties, and the faster oxidation yielding to a more difficult control over the oxide formation, the wet oxidation conditions can be considered as non-appropriated to develop gas-sensors with high performances.

To support the discussion, a new figure “Schematization of the oxidation processes of GaN-core NW under dry (a, b) and humid (c, d) atmosphere” has been added.

Comment 3:  What is effect of Ga2O3 thickness on the gas sensing performance? Authors should comment on this issue!

We have fabricated gas sensor devices integrating GaN/Ga2O3 core-shell NWs with the different thickness of Ga2O3, formed under dry and humid atmosphere. The devices are under testing and we are not sufficiently advanced in our measures to publish first results. They will be the subject of a new paper. We have added the following sentence in the conclusion:

By controlling the synthesis of GaN/β-Ga2O3 NWs, this paper constitutes a building block for developing high-efficient CO-gas sensors. GaN/β-Ga2O3 core-shell NW based sensor devices with different shell thickness (synthesis under dry and humid atmosphere) have been fabricated and are under investigation to establish the gas sensor performances as a function of the shell characteristics.

Comment 4:  The technical writing is not sound for example, In “High Resolution SEM images” H and R should be small, Energy Dispersive X-ray (EDX) written twice in the manuscript. Spelling mistakes (“developped” in the Conclusion), “This last decade”

We have corrected the technical writing and the spelling mistakes.

Comment 5:  Recent applications of GaN nanorods for low temperature gas sensors should be added to the introduction such as following papers.

1. Sensors and Actuators B 264, (2018) 353-362, https://www.sciencedirect.com/science/article/pii/S0925400518304969

2. Organic Electronics 65, 334-340 (2019), https://www.sciencedirect.com/science/article/pii/S1566119918306244

3. Dalton Transactions, 48, 1367-1375 (2019),

https://pubs.rsc.org/en/content/articlelanding/2019/dt/c8dt04709j#!divAbstract.

We have integrated these references in the manuscript.

The reference annotation has been adjusted by regarding the introduction of new ones proposed by reviewer or to support the added new discussions.

Reviewer 2 Report

Here are comments

1)      Introduction part of manuscript should be precise and impressive. English review may help improve the manuscript.

 2)      The detailed mechanism for the generation of GaN/Ga2O3 NWs is lacking.

 3)      Author provided future directions for GaN/β-Ga2O3 NWs as CO sensors. The utility of as synthesized GaN/Ga2O3 NWs can be illustrated to support the statements. Revealing the impact of interface modulated Ga2O3-shell thickness on CO sensors efficiency may provide an important finding of such research.

 4)      Figure quality can be modified.

Author Response

Manuscript ID: applsci-551370

Dear Editor,

We would like to thank each referee for his/her careful reading and remarks for our manuscript “GaN/Ga2O3 core/shell Nanowires growth: Towards high-efficient CO-sensors” by Q.C. Bui, et al. (manuscript ID: applsci-551370).

We have carefully considered each remark of the referees and have answered, modified, corrected and/or commented each one.

Please find below the referee’s comments immediately followed by our response. According to these comments, the manuscript has been revised, the text appearing in red corresponding to the added or modified informations.

All the modifications are appearing in red in the revised manuscript. With the reviewer’s comments, and the corresponding modifications, text adjustments have been performed, especially in the abstract, introduction and conclusion, to have a good coherence all along the manuscript. A new figure, as well as new references have been also added.

Thank you for your consideration.

Q.C. BUI and Dr. N. Gogneau

Reviewer 2:

Comment 1:  Introduction part of manuscript should be precise and impressive. English review may help improve the manuscript.

We have corrected the English and modify the organization of the introduction in order to be more precise. In order to avoid giving too more information in the introduction, which makes it confusing, we have removed the second paragraph dealing with the main criteria for a good sensor. We have also move the part dealing with the interest of the core-shell heterostructures for gas-sensors in order to be more impressive in our discussion.

Comment 2:  The detailed mechanism for the generation of GaN/Ga2O3 NWs is lacking.

We agree that this part of the discussion is missing in the original manuscript.  We have thus modified the discussion as follow:

Figures 2c and d establish the direct correlation between the diameter expansion and the oxidation conditions, with an evolution of the Ga2O3 shell thickness between 1 and 14 nm. Figure 2c shows the diameter expansion at 850°C (blue curve) and 900°C (green curve) as a function of the oxidation time, for O2 and N2 fluxes of 1.3 sccm and 5 sccm respectively and under dry atmosphere (Table 1, Series 1 and 3). The shell thickness increases with the time and the temperature, similarly to reported works in [50-55]. For the lower temperature (850°C), two different oxidation process can be distinguished: an initial rapid oxide-shell growth, followed by a second stage of reduced but continuous oxidation (after about 15 - 20 minutes of oxidation). For the higher temperature (900°C), the diameter expansion is more pronounced and can only be described by the first oxidation process, then demonstrating that the initial stage is dependent on the oxidation temperature. These descriptions of the oxidation process are consistent with the ones theoretically predicted [56] and experimentally observed [54-55, 57] in the literature. Then, the oxidation process is characterized by two different regimes: the interfacial reaction-controlled and the oxidation diffusion-controlled regimes. In interfacial reaction-controlled regime, the GaN oxidation process is controlled by the surface reaction rate, leading to a linear oxidation [58]. As the oxidation duration is important, as the oxide thickness increases. This process is also temperature dependent, as experimentally observed (Figure 2c). The increase of the oxidation rate with the temperature results from the increase of the O2 sticking probability attributed to a higher activation energy for O2 dissociation [54, 59]. In oxidation diffusion-controlled regime, the oxidation process is controlled by the reagent transport through the growing gallium oxide layer. The oxidation kinetic can thus be described by a parabolic model [58]. Although this regime is note observed for oxidation at 900°C (green curve), we can assume an under-linear thickening of the shell for higher oxidation times.

Comment 3:  Author provided future directions for GaN/β-Ga2O3 NWs as CO sensors. The utility of as synthesized GaN/Ga2O3 NWs can be illustrated to support the statements. Revealing the impact of interface modulated Ga2O3-shell thickness on CO sensors efficiency may provide an important finding of such research.

We have fabricated gas sensor devices integrating GaN/Ga2O3 core-shell NWs with the different thickness of Ga2O3, formed under dry and humid atmosphere. The devices are under testing and we are not sufficiently advanced in our measures to publish first results. They will be the subject of a new paper. We have added the following sentence in the conclusion:

By controlling the synthesis of GaN/β-Ga2O3 NWs, this paper constitutes a building block for developing high-efficient CO-gas sensors. GaN/β-Ga2O3 core-shell NW based sensor devices with different shell thickness (synthesis under dry and humid atmosphere) have been fabricated and are under investigation to establish the gas sensor performances as a function of the shell characteristics.

Comment 4:  Figure quality can be modified.

We have modified the figures in order to improve their quality and reading.

Round 2

Reviewer 1 Report

 The authors have made a proper revision for their manuscript.
The present form of the manuscript will be accepted.

Author Response

We would like to thank the referee for his/her careful reading and remarks for our manuscript “GaN/Ga2O3 core/shell Nanowires growth: Towards high response gas” by Q.C. Bui, et al. (manuscript ID: applsci-551370).

We have carefully considered the remarks and have answered, modified and corrected each one.

Please find below the referee’s comments immediately followed by our response. According to these comments, the manuscript has been revised using the "Track Changes" function in Microsoft Word.

Thank you for your consideration.

Q.C. BUI and Dr. N. Gogneau

The authors have made a proper revision for their manuscript. The present form of the manuscript will be accepted.

We thank the referee for this positive review.

Reviewer 2 Report

Comments

1)  Check the sentence "The shell thickness increases with the time and the temperature, similarly to reported works in [50-55]. "

2) Check the sentence (line number 420-421).

Authors are suggested to thoroughly read the manuscript and also check the figure captions. It can be accepted after minor revision.

Author Response

We would like to thank the referee for his/her careful reading and remarks for our manuscript “GaN/Ga2O3 core/shell Nanowires growth: Towards high response gas” by Q.C. Bui, et al. (manuscript ID: applsci-551370).

We have carefully considered the remarks and have answered, modified and corrected each one.

Please find below the referee’s comments immediately followed by our response. According to these comments, the manuscript has been revised using the "Track Changes" function in Microsoft Word.

Thank you for your consideration.

Q.C. BUI and Dr. N. Gogneau

Check the sentence "The shell thickness increases with the time and the temperature, similarly to reported works in [50-55]. "

We agree that the sentence gives the idea that similar oxidation of NWs have been reported, while the references concern the oxidation of films. We have thus modified the sentence as follow:

The shell thickness increases with the time and the temperature. This behavior is similar to the one observed for oxidized GaN or AlN films [50-55].

Check the sentence (line number 420-421).

In order to be clear, the sentence has been modified as follow:

At this stage, it is important to note that blue dots of oxygen atoms can be seen in the GaN-core area. However, this is not the signature of oxygen incorporation inside the GaN-core. Because the Ga2O3 shell surrounds the GaN core, during the EDX analyses, the electron beam is transmitted through the entire NWs (GaN plus Ga2O3), then causing an overlaps of the core and shell EDX signatures. The observed blue dots of oxygen in the core area is thus a visual effect specific to the NW heterostructure configuration as we have already described in [65].

The new reference has been added in the manuscript.

Authors are suggested to thoroughly read the manuscript and also check the figure captions. It can be accepted after minor revision.

We have carefully read the manuscript and checked the figure captions. The modifications are appearing in the manuscript using the "Track Changes" function in Microsoft Word.
